# Microbiological quality assessment of potential pathogenic bacteria and multidrug resistance patterns in commercial electrolyte drinks in Dhaka, Bangladesh

Sananda Saha[ID], Ayesha Maliha Khan, Fahim Kabir Monjurul Haque[ID]*

Department of Microbiology, School of Life Sciences, BRAC University, Pragati Sarani, Merul Badda, Dhaka, Bangladesh

* fahim.haque@bracu.ac.bd

## Abstract

Local electrolyte drinks in Bangladesh are produced without approval from the Bangladesh Standards and Testing Institution (BSTI). Hence, this study aimed to identify foodborne bacteria from electrolyte drinks and their antimicrobial resistance profiles to assess microbiological safety. A total of 40 electrolyte drink samples, representing 9 local brands in Dhaka, were evaluated for microbiological quality. Bacteria were present in all samples. Around 40% of the samples exceeded the aerobic plate count limit established by the Food and Drug Administration (FDA). Additionally, 55% of the samples tested positive for fecal contamination, surpassing the accepted limit set by the Food and Agriculture Organization of the United Nations (FAO). Both Gram-negative (*Klebsiella pneumoniae, Escherichia coli, Pseudomonas aeruginosa, Acinetobacter baumannii,* and *Vibrio* spp*.)* and Gram-positive (*Staphylococcus aureus, Staphylococcus epidermidis*, and *Listeria* spp*.)* bacteria were identified. Each sample contained more than one bacterial species. A total of 110 isolates from each culture-positive sample were screened for antimicrobial susceptibility. Multidrug resistance varied among all Gram-positive and Gram-negative bacterial species. The highest level of resistance was seen in broad-spectrum antibiotics (amoxicillin, ampicillin, ceftriaxone, cefotaxime, and cefepime). About 28% of Gram-positive and 19% of Gram-negative bacteria were hemolytic. Coagulase was found in 34% of the *Staphylococcus aureus* isolates. Moreover, 26% of the Gram-positive and 30% of the Gram-negative isolates were found to be resistant to human serum treatment. Such resistant bacteria found in popular electrolyte drinks thus represent a major health risk.

which permits unrestricted use, distribution, and reproduction in any medium, provided the original author and source are credited.

**Data availability statement:** All relevant data are within the manuscript and its Supporting Information files.

**Funding:** The author(s) received no specific funding for this work.

**Competing interests:** The authors have declared that no competing interests exist.

## Introduction

Electrolyte drinks are beverages that usually contain water and essential electrolytes such as sodium, potassium, magnesium, calcium, chloride, etc. These drinks are designed to restore the lost electrolytes during exercise, precipitation, or dehydration [1]. Electrolyte drinks, as opposed to ordinary water, are designed to promote healthy muscle and nerve function, increase fluid retention, and increase rehydration effectiveness. Global market demand has significantly increased because of their practical advantages and ease of use. Originally preferred by athletes, these drinks are now extensively consumed by the public and promoted as a means of rehydrating, reviving, and enhancing physical performance. As more individuals consume these products, it is essential to evaluate their overall quality, specifically regarding microbiological safety. This aspect is critical for maintaining the microbiological integrity and hygiene of food and beverages.

Drinks high in water, sugars, acids, and minerals create a conducive environment for microbial growth if improperly processed [2]. While electrolyte drinks are often made acidic to prevent this, contamination can still happen during manufacturing and distribution. Various sources, including raw ingredients, processing equipment, packaging, and air, can introduce harmful microbes. Compliance with safety regulations is critical to prevent health issues from these contaminants. Common microorganisms found in beverages include *Bacillus, Lactobacillus*, molds, and yeasts [3,4]. Poor sanitation or contamination during pasteurization can lead to high microbial loads that affect flavor, acidity, gas production, and potentially pose health risks. Thus, microbiological testing is vital to ensure product consistency and consumer safety.

Studies on the microbiological quality of commercially available electrolyte drinks are few to none. However, related research indicates that similar beverages, including energy drinks, soft drinks, and bottled water, often show contamination. Bottled water frequently hosts coliforms, *Staphylococcus, Escherichia coli*, and *Pseudomonas* [5,6]. Soft drinks are associated with bacteria such as *Clostridium, Gluconobacter*, and *Bacillus*, which are linked to spoilage [7]. Molds and yeasts are also prevalent in sports drinks [8]. These findings imply that inadequate sanitary practices and storage conditions can lead to microbial contamination in commercially produced beverages. Although various studies have explored the physiological effects of electrolyte drinks on athletes, none have specifically addressed their microbiological safety or quality [9].

Research on electrolyte drinks reveals a gap in knowledge regarding their microbiological quality. Unlike other beverages, these drinks contain varying levels of carbohydrates and salts that can affect microbial viability, particularly in formulations lacking preservatives. Their mineral and sugar content, along with nearly neutral pH levels, may foster microbial growth, raising safety concerns, especially for vulnerable populations. It is unknown whether these items comply with food safety regulations established by authorities such as the U.S. Food and Drug Administration (FDA) and the World Health Organization (WHO), due to the lack of studies on the subject. The FDA has established certain guidelines for sports drinks and energy drinks in the US [10]. The Ministry of Public Health, Thailand, has an unofficial recommendation for electrolyte drinks [11]; however, it seems those are frequently overlooked by the regulators.

Even small microbial contamination in electrolyte drinks can significantly impact safety, brand reputation, and customer trust. Spoilage organisms might not cause immediate illnesses but can lead to severe foodborne diseases like those caused by *Salmonella, E. coli*, or *Staphylococcus aureus*. To assess safety, microbial testing methods such as Aerobic Plate Count (APC) and coliform detection are employed. Particularly in hot, humid regions like Bangladesh, the potential for microbial growth in packaged drinks is heightened, yet there is a lack of regulatory oversight and limited research on these products. This scarcity of data makes it difficult to evaluate the health risks or compliance with global safety standards for electrolyte beverages in Bangladesh.

With all this in mind, this study set out to check how safe local electrolyte drinks sold in Dhaka really are by looking at which harmful microbes are present and seeing if those microbes are resistant to antibiotics. Since these drinks are produced in batches, the study made sure to buy samples from different batches to avoid repeats. Researchers also confirmed that each sample was still within its expiration date and that the bottles were sealed. Altogether, 40 electrolyte drinks from 9 different brands were collected from local stores across Dhaka and tested for their microbiological quality. To ascertain conformity and safety, the outcomes will be contrasted with the current microbiological standards established by regulatory entities. This study aims to fill the research gap by offering useful information to the field of food microbiology as well as knowledge that could help authorities, customers, and producers ensure the microbiological quality of electrolyte beverages on the marketplace.

## Materials and methods

### Sample collection

A total of 40 batch-wise electrolyte drink samples were collected from the local retail stores, ensuring proper seal and expiration date. Following the collection, every sample was sent promptly to the lab for a thorough microbiological examination.

### Sample processing

After the samples were filtered by using the membrane filtration method using a 0.45 μm filter paper, the filter paper was put into Falcon tubes with 30 ml of enrichment broth. Tryptic Soya Broth (TSB) and Alkaline Peptone (AP) water were used for enrichment, and the tubes were kept in an incubator for 24 hours at 37°C. Serial dilutions were carried out using a sterile sodium chloride solution. Later, microbiological tests were used to determine the amount and presence of microbial contaminants. All microbiological analyses were performed in triplicate, and results are presented as the mean of three independent experiments.

### APC

The spread plate method was used to perform aerobic plate count on Nutrient Agar (NA). 0.1 ml of the diluted solution to get single countable colonies was spread on the agar plate and incubated at 37°C for 24 hours. For calculating APC, the FDA's guideline was followed [12]. However, as electrolyte drinks did not have any separate protocols for APC, we followed the protocols for juice samples [13].

### Bacteria culture & identification

To identify the targeted bacteria, 0.1 ml (100 μl) of the diluted sample was spread on different selective media and incubated at 37 °C for 24 h. Following incubation, colony morphology and Gram staining were used to make the primary classification. S1 Table lists the various selective media that have been used, along with the colony morphology.

### DNA extraction and Polymerase Chain Reaction (PCR) amplification

A single colony from each plate was selected and sub-cultured into NA and incubated at 37°C for 24 hours. To create a uniform suspension, a bacterial colony was injected into 400 μL of 1X TE buffer in a sterile Eppendorf tube and vortexed. Following a 10-minute centrifugation at 13,000 rpm and 25°C, the pellet was resuspended in 400 μL of TE buffer and vortexed once more after the supernatant was extracted. To release DNA and break down cell membranes, the cells were heated to 100°C for 10 minutes. Following 5 minutes of cooling, a second centrifugation was carried out for 10 minutes at 13,000 rpm. Finally, 200 μL of the clear supernatant containing genomic DNA was moved to a new tube for molecular analysis. DNA samples were stored at −20°C.

To identify each bacterial sample at the molecular level, a 13 μL PCR reaction mixture was prepared. This included 6 μL of commercially prepared PCR Master Mix (with Taq DNA polymerase, dNTPs, magnesium chloride, and reaction buffer), 3 μL of nuclease-free water, 1 μL each of forward and reverse primers, and 2 μL of extracted bacterial DNA. Details about the primers, PCR thermocycler settings, and amplicon size are provided in Table 1. The PCR products were checked by running them on a 1.7–2% agarose gel with 1X TAE buffer, staining with ethidium bromide, and visualizing under a UV transilluminator.

### Antimicrobial Susceptibility Test (AST)

According to the standards given by the Clinical and Laboratory Standards Institute (CLSI), published in 2024 [29] and for *Vibrio* spp. CLSI 2015 guidelines [30], the testing was conducted using the standardized Kirby-Bauer disk diffusion method. Before lawning the bacterial solution onto Muller-Hinton Agar (MHA) plates, a 0.5 McFarland standard suspension was created using overnight cultures. Sterile placement of antibiotic discs onto the plates was done. After 24 hours of incubation at 37 °C, the plates were examined to determine the antibiotics' zone of inhibition. The CLSI-defined breakpoints for each antibiotic-bacterium combination were used to interpret these measurements as Sensitive (S), Intermediate (I), or Resistant (R). S2 Table lists the antibiotics tested along with their class, efficacy, disc concentration, and interpretation parameters.

### Multidrug Resistance (MDR)

Resistance to at least one agent in three or more antimicrobial classes is considered as MDR [31].

### Multiple-Antibiotic Resistance (MAR) index

The MAR index was calculated as a ratio of resistance to the number of antibiotics by the isolates ('a') to that of the number of antibiotics used ('b') [32]. A MAR index value of ≤0.2 indicates that the isolate originated from a source with low antibiotic exposure, while a value greater than 0.2 implies that the isolate originated from a high-risk source where antibiotics are routinely used.

### Antimicrobial Resistance (AMR) Genes

Our objective was to figure out whether the isolates carried any known AMR genes based on the AST results. The presence of β-lactamase and carbapenemase-associated genes, such as $bla_{CTX-M}$, $bla_{TEM}$, $bla_{SHV}$, $bla_{NDM}$, $bla_{VIM}$, and $bla_{KPC}$, were screened in Gram-negative isolates and Methicillin-Resistant Staphylococcus aureus (MRSA) gene, *mecA* was screened in *S. aureus* isolates using PCR.

### Hemolysis

The ability of the bacterial isolates to lyse red blood cells and break down hemoglobin was assessed using a blood agar plate. A single colony was streaked on the blood agar plate and incubated at 37°C for 24 hours, and later a zone of

**Table 1. Primers and PCR conditions used in this study.**

| Organism/ Primer Name | Primer Sequence | Targeted gene | Product Size | PCR Condition | Reference |
|---|---|---|---|---|---|
| *Klebsiella pneumoniae* | KP Pf-F: 5′-ATTTGAAGAGGTTGCAACGAT-3′) KP Pf R:5′-TTCACTCTG AAGTTTTTTGTGTTC-3′ | 16S–23S rRNA | 130 bp | The cycling conditions were 10 min at 94 °C, followed by 35 cycles of 30 s at 94 °C, 20 s at 57 °C, and 20 s at 72 °C, followed by a 10 min hold at 72 °C. | [14] |
| *Escherichia coli* | ECO-F: GACCTCGGTTTAGTTCACAGA ECO-R: CACACGCTGACGCTGACCA | *Gnd* | 585 bp | Initial denaturation at 95°C for 5 min; 35 cycles of denaturation at 94°C for 45s, annealing at 45°C for 45s, and extension for 1 min, followed by a final extension at 72°C for 5 min. | [15] |
| *Staphylococcus aureus* | NUC F: 5′ - GCGATTGATGGTGATACGGT T-3′ NUC R: 5′- AGCCAAGCCTTGACG AACTAAAGC-3′ | *nuc* gene | 279 bp | Initial denaturation at 95°C for 5 min, followed by 30 cycles of denaturation at 95 °C for 1 min, an annealing temperature of primers was 55°C for 45 sec, and extension at 72°C for 1 min. The final extension was conducted at 72°C for 10 min | [16] |
| *Staphylococcus epidermidis* | gseA F: ATGAAAAAGAGATTTTTATCT gseA R:GTTTGGTGACACTCTTAAG | *gseA* | 503 bp | An initial denaturation at 94 °C for 2 min, followed by 30 cycles for amplification, consisting of 94 °C for 10 s, 50 °C for 10 s, and 72 °C for 60 s, followed by a final extension step at 72 °C for 4 min. | [17] |
| *Listeria* spp. | prs_F: 5′- GCTGAAGAGATTGCGAAAGAAG −3′ prs_R: 5′- CAAAGAAACCTTGGATTTGCGG −3′ | *prs* | 370 bp | 94°C for 3 min; 35 cycles of 94°C for 0.40 min, 53°C for 1.15 min, and 72°C for 1.15 min; and one final cycle of 72°C for 7 min in a thermocycler | [18] |
| *Pseudomonas aeruginosa* | PA-SS F: 5-GGGGGATCTTCGGACCTCA-3 PA-SS R: 5-TCCTTAGAGTGCCCACCCG-3 | 16S rRNA | 956 bp | Initial denaturization for 2 min at 95°C, 25 cycles were completed, each consisting of 20 s at 94°C, 20 s at the annealing temperature of 58°C, and 40 s at 72°C. A final extension of 1 min at 72°C was applied. | [19] |
| *Acinetobacter baumannii* | blaOXA51 F: 5′ TAATGCTTTGATCGGCCTTG 3′ blaOXA51 R: 5′ TGGATTGCACTTCATCTTGG 3′ | blaOXA-51 | 353 bp | Initial Denaturation at 94°C for 5 min, followed by 30 cycles of 94 °C for 1 min, 55 °C for 1 min, 72 °C for 1 min, and a final extension at 72 °C for 10 min. | [20] |
| *Vibrio* spp. | vibrio_genus F: 5′- GTCARATTGAAAARCART-TYGGTAAAGG 3′ vibrio_genus R: 5′ -ACYTTRATRCG NGTTTCRTTRCC- 3′ | 16S rRNA | ~ (620–689) bp | Initial denaturation at 94 °C for 5 min, followed by 25 cycles of 94 °C for 30 s, 60 °C for 30 s, 72 °C for 30 s, finishing with a final extension at 72 °C for 10 min | [21] |
| mecA | mecA P4 2821 F: 5′– TCCAGATTAC AACTTCACCAGG – 3′ mecA P7 2822 R: 5′ – CCACTTCATATCTTGT AACG – 3′ | *mecA* | 162 bp | Initial denaturation at 95°C for 3 min, followed by 40 cycles of DNA denaturation at 94 °C for 60 s, primer annealing at 55 °C for 30 s, and DNA extension at 72 °C for 90 s. After the final cycle, final extension at 72 °C for 3.5 min. | [22] |
| blaCTX-M | F: 5′ -ACGCTGTTGTTAGGAAGTG −3′ R: 5′ -TTGAGGCTGGGTGAAGT – 3′ | $bla_{CTX-M}$ | 857 bp | The PCR amplification comprised an initial denaturation of 3 min at 94°C, followed by 36 cycles of denaturation at 94°C for 1 min, annealing at 58°C for 30 s, extension at 72°C for 1 min, and a final extension of 10 min at 72°C. | [23] |

*(Continued)*

**Table 1.** (Continued)

| Organism/ Primer Name | Primer Sequence | Targeted gene | Product Size | PCR Condition | Reference |
|---|---|---|---|---|---|
| blaTEM | TEM-F:5'-AAAATTCTTGAAGACG-3' TEM-R:5'-TTACCAATGCTTAATCA-3' | $bla_{TEM}$ | 1080 bp | Initial denaturation at 94°C for 3 min; 35 cycles of denaturation at 94°C for 30s, annealing at 50°C for 30s, and extension at 72°C for 2 min, followed by a final extension at 72°C for 10 min. | [24] |
| blaSHV | F: 5' -TACCATGAGCGATAACAGCG- 3' R: 5' -GATTTGCTGATTTCGCTCGG- 3' | $bla_{SHV}$ | 450 bp | Amplification involved an initial 5-min step at 95°C, followed by 32 cycles of 1 min at 94°C, 40 s at 55°C, and 40 s at 72°C, concluding with a final extension of 5 min at 72°C. | [25] |
| blaKPC | KPC-F:5'-CATTCAAGGG CTTTCTTGCTGC-3' KPC-R:5'-ACGACGGCATAGTC ATTTGC-3' | $bla_{KPC}$ | 498 bp | Initial denaturation at 95°C for 5 min; 35 cycles of denaturation at 95°C for 30s, annealing at 57°C for 30s, and extension at 72°C for 40s, followed by a final extension at 72°C for 10 min. | [26] |
| blaNDM | F: GGTTTGGCGATCTGGTTTTC R: CGGAATGGCTCATCACGATC | $bla_{NDM}$ | 621 bp | Initial denaturation at 94°C for 10 min; 36 cycles of denaturation at 94°C for 30s, annealing at 52°C for 40s, and extension at 72°C for 50s, followed by a final extension at 72°C for 5 min. | [27] |
| blaVIM | VIM-F:5'-GGTGTTTGGTCGCATAT CGCAA-3' VIM-R:5'-ATTCAGCCAGATCGGC ATCGGC-3' | $bla_{VIM}$ | 501 bp | Initial denaturation at 95°C for 5 min; 35 cycles of denaturation at 95°C for 45s, annealing at 60°C for 45s, and extension at 72°C for 45s, followed by a final extension at 72°C for 10 min. | [28] |

hemolysis was observed. In the blood agar, the isolates with hemolysins produced either a clear zone (beta hemolysis) or a partial clear zone (alpha hemolysis). No clear zone indicated no hemolysis of red blood cells (gamma hemolysis). This test was performed to identify the pathogenic bacteria from the isolated bacteria [33].

### Coagulase test

Raw rabbit plasma was diluted using Phosphate Buffered Saline (PBS). For the tube coagulase test, in 0.5 ml of diluted rabbit plasma, 2–3 single colonies of bacteria were collected and inoculated. The tubes were incubated at 37°C for 4 hours and observed for clot formation. Any clot formation is counted as a positive result. If no clot formation was observed, then the tubes were incubated for 24 hours. If there was no clot formation at the end of 24 hours, it was considered a negative result. This test was performed to differentiate between *Staphylococcus epidermidis* and pathogenic *S. aureus* [34].

### Serum resistance assay

A serum resistance assay was performed using a modified version of the main protocol [35,36]. A single colony grown overnight on a fresh NA plate was transferred into 1 ml of fresh LB and incubated in a shaker incubator at 37°C for 2 hours. The culture was centrifuged at 8000 x g for 7 minutes, and the pellet was resuspended in 1 ml of sterile sodium chloride saline solution after removing the supernatant. In a sterile 96-well microtiter plate, 20 μl of the culture solution was mixed with 180 μl of pooled normal human serum (NHS). Serial tenfold dilutions were prepared up to $10^{-7}$. From each dilution, 10 μl was plated on an NA plate to represent the 0-hour time point. The microtiter plate was then incubated at 37°C for 3 hours, and 10 μl from each dilution was again plated on an NA plate to represent the 3-hour time point. Both

sets of plates were incubated at 37°C for 24 hours. Colony-forming units (CFU) were counted at the start (0 h) and end (3 h) of the incubation period, and a paired t-test was used to assess serum resistance. Isolates were considered resistant if the CFU count increased or remained the same after three hours, and sensitive if the count decreased. Each serum assay experiment was performed in triplicate for each isolate [37].

## Results

### Fecal contamination

*E. coli* or any other fecal coliform must not be present in any electrolyte drink sample [11]. Hence, the coliform count was not taken. Only the presence of fecal coliform (blue colonies on M-Fc Agar) was enough to consider the sample as contaminated. Out of 40 samples, 55% (22 samples) were positive for fecal contamination.

### APC

After enrichment and serial dilution, 0.1 ml of each sample was spread on a NA plate to obtain the aerobic plate count. Among 40 samples, the APC of 16 samples exceeded the limited contamination level of the FDA [12]. Table 2 shows that 40% of the samples exceeded the limited threshold.

From electrolyte drinks, both Gram-positive and Gram-negative microbes were found. The Gram-positive organisms that were found were *S. aureus, S. epidermidis,* and *Listeria* spp*.* The identified Gram-negative species included *Vibrio* spp*., Acinetobacter baumannii, Pseudomonas aeruginosa, E. coli,* and *Klebsiella pneumoniae*. Gram staining was used to do the primary identification (S1–S8 Figs). PCR was used to confirm each isolate (S9–S16 Figs).

### Detection of multiple bacteria

More than one bacterial species was also found in every individual sample. Fig 1 shows that three different types of bacterial species were found in most of the samples. Only one sample had seven of the targeted bacterial species, whereas samples with five or six bacterial types were detected less frequently. All eight of the targeted organisms were not detected in any samples.

### AST

For this study, from each culture-positive sample, one isolate was randomly selected to identify the antimicrobial susceptibility pattern of the different organisms. A total of 110 isolates were selected and subjected to the Kirby-Bauer Disc Diffusion method. The total number of isolates for which antibiotic susceptibility was examined was *K. pneumoniae* 26 isolates, *E. coli* 11 isolates, *P. aeruginosa* 8 isolates, *A. baumannii* 6 isolates, *Vibrio* spp. 10 isolates, *S. aureus* 15 isolates, *S. epidermidis* 16 isolates, and *Listeria* spp. 18 isolates.

The broad-spectrum antibiotics that showed the highest level of resistance were amoxicillin (77.14%), ampicillin (70%), ceftriaxone (58.82%), cefotaxime (50.98%), and cefepime (48.18%). The narrow-spectrum antibiotics for Gram-positive

**Table 2. Total aerobic plate count result.**

| No of Electrolyte Samples | Limited threshold | Microbial load in CFU/ml (Number of the total samples contaminated) | | |
|---|---|---|---|---|
| 40 | $10^2$-$10^5$ CFU/mL | $<10^2$ (10) | $10^2$-$10^5$ (14) | $>10^5$ (16) |

The total APC results for all the electrolyte drinks are mentioned, along with the limited threshold of contamination level allowed. The number of total samples contaminated is given in the parentheses.

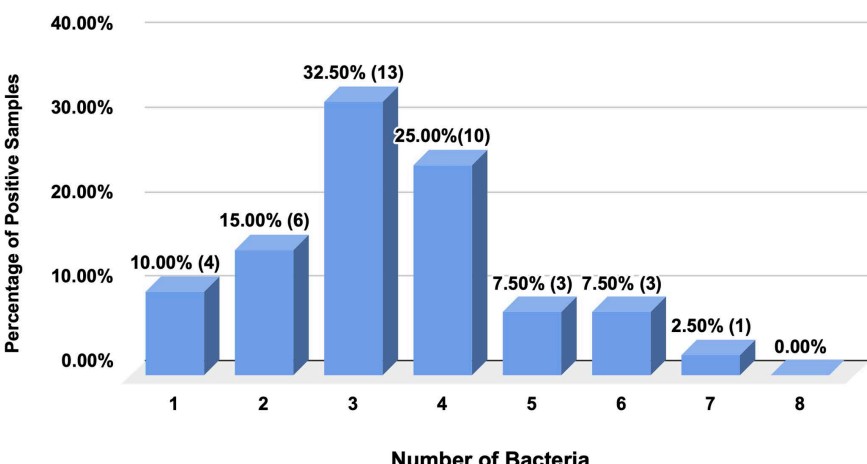

**Fig 1. Detection of multiple bacteria in samples.** This figure illustrates the percentage of samples in which more than one bacterial species was found. The number of samples is given in parentheses, along with the percentage.

isolates, both vancomycin (14.29%) and linezolid (16.33%), showed almost similar levels of resistance. The antibiotic resistance patterns of all 110 isolates are given in Fig 2.

In comparison to Gram-positive bacteria, Gram-negative bacteria showed a higher incidence of resistance (Fig 3). High levels of resistance to penicillin and third-generation cephalosporins, such as ampicillin, amoxicillin, cefotaxime, and ceftriaxone, were demonstrated by *K. pneumoniae* and *E. coli*. These species were also frequently shown to be resistant to tetracycline, ciprofloxacin, and trimethoprim–sulfamethoxazole. *A. baumannii* and *P. aeruginosa* both showed significant multidrug resistance, with high levels of resistance to several antibiotic classes, such as carbapenems, aminoglycosides, cephalosporins, and fluoroquinolones. On the other hand, Gram-positive bacteria such as *S. aureus, S. epidermidis,* and *Listeria* spp. showed lower overall resistance rates. Vancomycin and linezolid showed modest resistance rates, but resistance to macrolides, tetracyclines, and certain β-lactams was the most common resistance seen in staphylococci. All the tested drugs showed low to moderate resistance in *Listeria* spp. Although less sensitivity was noted in *K. pneumoniae* and *A. baumannii*, suggesting growing resistance to last-resort antibiotics, carbapenems and tigecycline were amongst the most effective drugs across all organisms. The antibiotic resistance patterns of all isolates are given in Fig 3 as a heatmap. Additionally, all isolates' comprehensive resistance profiles, as well as intermediate and sensitive data, are included in the S3 Table.

## MDR

Multidrug resistance was present in about 79.09% of all isolates. The 21 antibiotics used for this study fall under 12 antibiotic classes. Nearly most of both Gram-negative and Gram-positive isolates showed resistance to more than three classes of antibiotics. The highest level of resistance was observed in 5 classes of antibiotics (Fig 4).

## MAR index

The number of antibiotics to which an isolate is resistant, divided by the total number of antibiotics tested, is the MAR index. Based on the results, the isolates' MAR indices differ significantly (Table 3). The MAR index ranged from 0.26 to 0.5 for 36.37% of the isolates. A MAR value of less than 0.2 was present in almost 32% of the isolates, suggesting sources of high-risk contamination. A MAR value greater than 0.75 was present in almost 9.09% of the isolates.

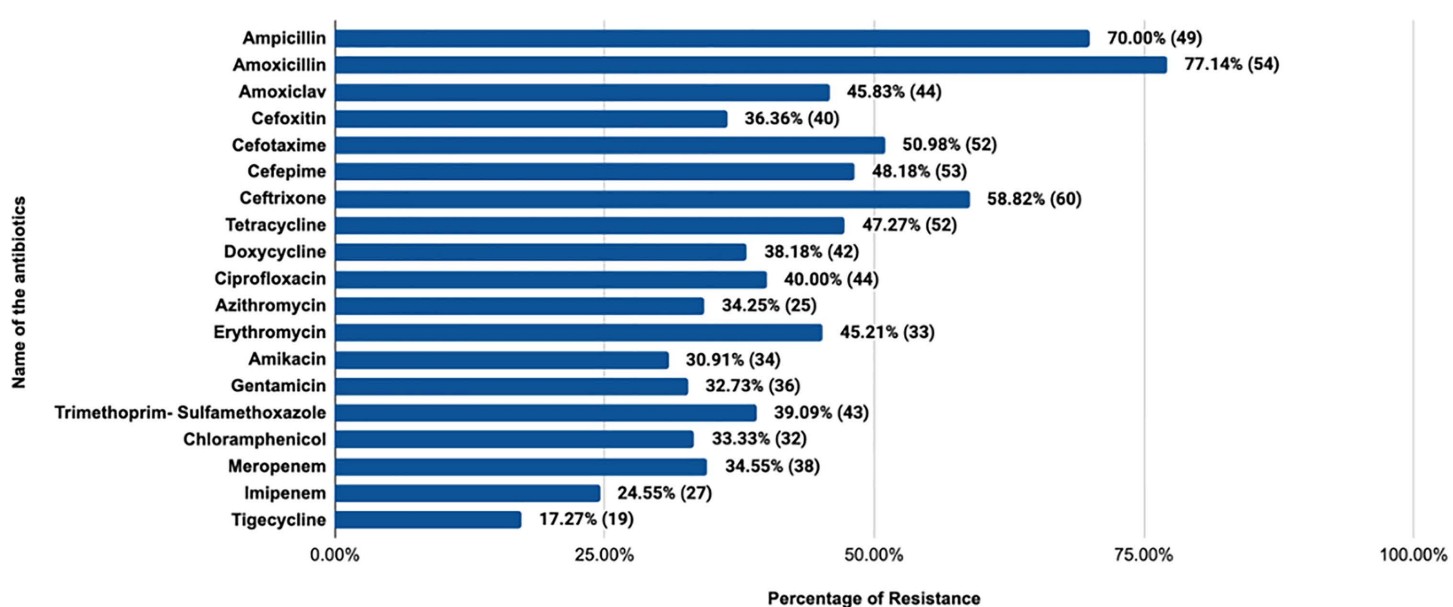

## a) Resistance Observed Against Broad-Spectrum Antibiotics

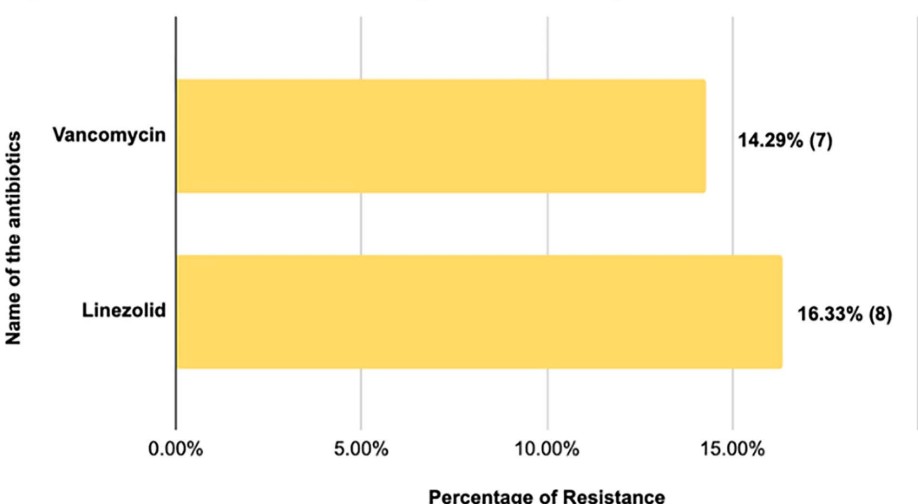

## b) Resistance Observed Against Gram-positive Antibiotics

**Fig 2. Resistance percentage found in each isolate.** The number of isolates that exhibited antibiotic resistance is indicated in parentheses on the right side of each bar. The isolates with intrinsic resistance were not included in the percentage analysis. **a)** Resistance observed against broad spectrum antibiotics consisting of 110 isolates **b)** Resistance observed against Gram-positive antibiotics consisting of 31 isolates of *S. aureus* and *S. epidermidis*.

### AMR genes

Based on the phenotypic AST result, we targeted known genes responsible for resistance mechanism. PCR was used to study the prevalence of AMR genes within the bacterial isolates (S17–S23 Figs). The presence of β-lactamase and

## Heatmap of Phenotypical Antimicrobial Resistance Among Bacterial Isolates

**Fig 3. Heatmap of phenotypical antimicrobial resistance among bacterial isolates.** Here, the blank cells indicate antibiotics that are not tested or not recommended due to intrinsic resistance. Percentages were calculated based on the total number of isolates tested for each organism.

carbapenemase-associated genes, such as $bla_{CTX-M}$, $bla_{TEM}$, $bla_{SHV}$, $bla_{NDM}$, $bla_{VIM}$, and $bla_{KPC}$, was confirmed in isolates of *K. pneumoniae, E. coli, P. aeruginosa, A. baumannii,* and *Vibrio* spp. For MRSA the primary target gene was *mecA*. Among *S. aureus* isolates, the *mecA* gene was detected in 20.00% of isolates, indicating the presence of MRSA. Extended-spectrum β-lactamase (ESBL) genes were shown to be highly prevalent in *K. pneumoniae*. 76.92% of isolates had the $bla_{CTX-M}$ gene, followed by $bla_{TEM}$ (57.69%) and $bla_{SHV}$ (46.15%). While $bla_{KPC}$ was not found, the carbapenemase genes $bla_{NDM}$ and $bla_{VIM}$ were found in 7.69% and 46.15% of isolates, respectively. $Bla_{TEM}$ was found in 63.64% of *E. coli* isolates, but $bla_{CTX-M}$ and $bla_{SHV}$ were not found Table 4. In 9.09% and 27.27% of isolates, respectively, the carbapenemase genes $bla_{NDM}$ and $bla_{VIM}$ were found. The $bla_{KPC}$ gene was not found. *P. aeruginosa* isolates did not have any of the studied AMR genes ($bla_{NDM}$, $bla_{VIM}$, or $bla_{KPC}$). Carbapenemase-associated genes were commonly found in *A. baumannii*, with $bla_{VIM}$ and $bla_{NDM}$ found in 50.00% and 33.33% of isolates, respectively. Neither $bla_{KPC}$ nor ESBL genes were found. The carbapenemase-associated gene $bla_{VIM}$ was found in 40% of isolates of *Vibrio* spp. In the present study, neither $bla_{NDM}$ nor $bla_{KPC}$ were found.

## Hemolysis patterns

The hemolysis test was performed to check whether the isolated organism has the pathogenic ability to lyse red blood cells. Table 5 shows the percentage of hemolysis patterns.

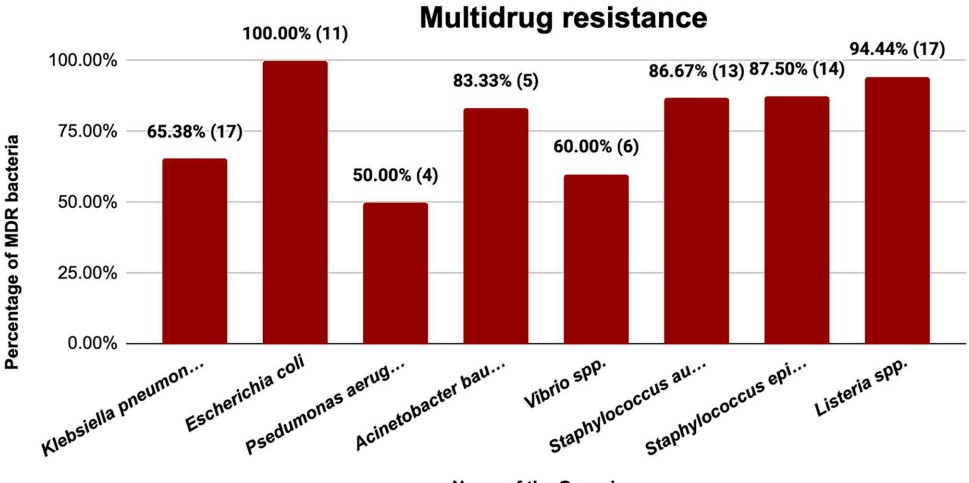

**Fig 4. Multidrug resistance.** The number of MDR isolates for each organism is given in parentheses, along with the percentage.

**Table 3. Multiple-antibiotic resistance (MAR) index.**

|  | Isolates | MAR index < 0.25 | MAR index = (0.26–0.5) | MAR index = (0.6–0.75) | MAR index > 0.75 |
|---|---|---|---|---|---|
| **Total** | 110 | 35 (31.82%) | 40 (36.37%) | 25 (22.28%) | 10 (9.09%) |
| **Klebsiella pneumoniae** | 26 | 11 | 4 | 6 | 5 |
| **Escherichia coli** | 11 | 3 | 4 | 3 | 1 |
| **Pseudomonas aeruginosa** | 8 | 4 | 0 | 2 | 2 |
| **Acinetobacter baumannii** | 6 | 2 | 1 | 1 | 2 |
| **Vibrio spp.** | 10 | 5 | 4 | 1 | 0 |
| **Staphylococcus aureus** | 15 | 2 | 8 | 5 | 0 |
| **Staphylococcus epidermidis** | 16 | 3 | 9 | 4 | 0 |
| **Listeria spp.** | 18 | 5 | 10 | 3 | 0 |

MAR index of all the different types of bacteria in electrolyte drink samples.

## Coagulase test

*S. aureus* was the most prevalent skin pathogen among all the organisms targeted. Therefore, additional screening was performed to evaluate its pathogenic potential. In the coagulase test, the plasma coagulase-reacting factor was detected in 33.33% of the isolates.

## Serum resistance assay

The purpose was to determine whether the isolates would survive serum treatment. The 110 isolates from 8 targeted organisms were tested against pooled human serum. Findings show that (Table 6) *Vibrio* spp. isolates were the most

**Table 4. Detection of antimicrobial resistance genes.**

| Primer Name | Organism Name | | | | | |
|---|---|---|---|---|---|---|
| | *Staphylococcus aureus* | *Klebsiella pneumoniae* | *Escherichia coli* | *Pseudomonas aeruginosa* | *Acinetobacter baumannii* | *Vibrio spp.* |
| **mecA** | 20.00% | -- | -- | -- | -- | -- |
| **blaCTX-M** | -- | 76.92% | 0.00% | -- | -- | -- |
| **blaTEM** | -- | 57.69% | 63.64% | -- | -- | -- |
| **blaSHV** | -- | 46.15% | 0.00% | -- | -- | -- |
| **blaNDM** | -- | 7.69% | 9.09% | 0.00% | 33.33% | 0.00% |
| **blaVIM** | -- | 46.15% | 27.27% | 0.00% | 50.00% | 40.00% |
| **blaKPC** | -- | 0.00% | 0.00% | 0.00% | 0.00% | 0.00% |

AMR gene distribution among bacterial isolates recovered from electrolyte drink samples, as determined by PCR. Percentages indicate the proportion of isolates positive for each gene within a given species. "--" denotes that the gene was not tested or not applicable for that organism.

**Table 5. Hemolysis patterns.**

| Organism Type | Name of the Organism | Percentage of Hemolysis Type | | |
|---|---|---|---|---|
| | | **Alpha** | **Beta** | **Gamma** |
| **Gram-negative** | *Klebsiella pneumoniae* (26) | 0 | 7.7 | 92.3 |
| | *Escherichia coli* (11) | 0 | 72.72 | 27.27 |
| | *Pseudomonas aeruginosa* (8) | 0 | 62.5 | 37.5 |
| | *Acinetobacter baumannii* (6) | 0 | 0 | 100 |
| | *Vibrio* spp. (10) | 10 | 60 | 30 |
| **Gram-positive** | *Staphylococcus aureus* (15) | 0 | 40 | 60 |
| | *Staphylococcus epidermidis* (16) | 0 | 62.5 | 37.5 |
| | *Listeria* spp. (18) | 11.2 | 66.7 | 22.23 |

Hemolytic pattern of the isolated organisms. The number of isolates for each organism is given in parentheses beside the organism's name.

**Table 6. Serum resistance assay results.**

| Organism | Percentage of Resistance level |
|---|---|
| *Staphylococcus aureus* (15) | 60.00 |
| *Staphylococcus epidermidis* (16) | 50.00 |
| *Listeria* spp. (18) | 61.11 |
| *Klebsiella pneumoniae* (26) | 57.69 |
| *Escherichia coli* (11) | 54.55 |
| *Pseudomonas aeruginosa* (8) | 50.00 |
| *Acinetobacter baumannii (6)* | 16.67 |
| *Vibrio* spp. (10) | 70.00 |

Percentage of resistance level in the serum resistance assay of all 110 isolates. The number of isolates for each organism is given in parentheses beside the organism's name.

resistant to human serum at a rate of 70% (out of 10 *Vibrio* spp. isolates), followed by *Listeria* spp. (18 isolates) and *S. aureus* (15 isolates) at 61.11% and 60%, respectively.

## Discussion

Electrolyte beverages help muscles and nerve's function, replenish depleted ions, and promote fluid retention. They were once primarily promoted to athletes, but because the general population increasingly consumes them, evaluating their microbiological quality is crucial for public health. Electrolyte beverages are still mostly overlooked, despite research on the microbiological quality alcoholic beverages, bottled water, sports drinks, and energy drinks. To fill this gap, we collected samples of electrolyte drinks from retail stores in Dhaka, Bangladesh, and tested them in the lab. To the best of our knowledge, this report is the first to assess the microbiological quality of these beverages in Bangladesh as well as globally.

In the current study, the microbiological quality of electrolyte drinks was assessed through several parameters, such as APC, fecal contamination, AST, MDR, MAR index, AMR genes, hemolysis, coagulase, and serum resistance. The results revealed that all samples included bacteria, with 55% testing positive for fecal contamination and 25% exceeding the Food and Agricultural Organization's (FAO) permissible limit for APC. There were several types of not only Gram-positive bacteria, but also Gram-negative bacteria found in each sample, with MDR bacteria, especially to meropenem and azithromycin in overall 110 isolates. 34% of *S. aureus* isolates tested positive for plasma CRP, whereas 28% of Gram-positive and 19% of Gram-negative bacteria had hemolysis. Most hemolytic isolates showed beta-hemolysis, which is alarming because these bacteria's endotoxins can harm red blood cells and hemoglobin [38]. Furthermore, human serum was resisted by 26% of Gram-positive and 30% of Gram-negative isolates, suggesting that these drinks could be a significant health concern.

The APC count in our study showed that 60% of the sample contained an acceptable amount of microbial load, ranging from $10^2$–$10^5$ CFU/ml, comparable to fruit juice and bottled water studies [4]. However, the rest 40% of the samples exceeded the limited accepted standard set by the FDA (Philippines), indicating high microbial load reported in related beverage products [39,40]. More than half (55%) of the samples in our study were positive for fecal contamination shown by the identification of *E. coli* or other coliform, indicating possible water contamination or improper handling techniques. According to the WHO and FDA guidelines, fecal coliform must be absent from 100 ml of water samples or, in this case, electrolyte drinks where water is used as a raw material. Therefore, the product will be identified as unsafe even if a small bacterial load of the previously specified microorganisms is found. These results line up with what's been found in studies on carbonated soft drinks in Bangladesh and bottled water in Trinidad [41,42]. While there's plenty of research on how electrolyte drinks affect rehydration and sports performance [43,44], very few studies have looked at their microbiological safety, underscoring the need for thorough contamination screening. The factory atmosphere, machinery, storage, and raw materials like water are only a few of the production stages where contamination can happen [45]. While electrolytes and carbohydrates may promote microbial growth, filtration errors can bring in heterotrophic bacteria or fecal coliforms. The elevated microbial loads and fecal coliform abundance in our samples indicate that there was a large danger of cross-contamination from raw materials, packaging, or handling [46,47].

Microbial survival is impacted by the low pH (3.5–4.5) and fluctuating sugar content of electrolyte drinks [48]. Acid-tolerant organisms such yeasts, *Lactobacillus,* and *Staphylococcus* thrive in their acidic environment [49–51]. 78% of our samples tested positive for *S. aureus* or *S. epidermidis*, which is in line with earlier observations [52,53]. Furthermore, at high concentrations, sodium and potassium salts can inhibit growth, but at lower concentrations, they may improve stress tolerance. For instance, *S. aureus* can flourish at high NaCl concentrations when minerals are present [54], and comparable adaptation mechanisms have been documented in *S. aureus* and *E. coli* [55–57].

Alarming resistance levels were found in the AST results (Fig 2). While sensitivity to tigecycline and meropenem remained high, the highest resistance was found against amoxicillin (77.14%), ampicillin (70%), ceftriaxone (58.82%), cefotaxime

(50.98%), cefepime (48.18%), tetracycline (47.27%), and amoxiclav (45.83%). Vancomycin (14.29%), linezolid (16.33%), tigecycline (17.27%), imipenem (24.55%), amikacin (30.91%), gentamicin (32.73%), chloramphenicol (33.33%), and meropenem (34.55%) showed the least amount of resistance. These results are consistent with reports of fourth-generation cephalosporin resistance in Dhaka and amoxiclav-resistant bacteria in Ghanaian drinking water. International studies have also found penicillin and extended-spectrum cephalosporin resistance in waterborne *Enterobacteriaceae* [58–61].

Around 80% of the total isolates were MDR. 100% of the *E. coli* isolates were found to be MDR. Conversely, *Listeria* spp. had an MDR rate of 94.44%, followed by *S. epidermidis* at 87.50%. These results are consistent with reports of *E. coli* (36.89%) and *Staphylococcus* spp. (18.9%) in fruit juices from Bangladesh [62], as well as environmental water and household supply investigations in Bangladesh that reveal MDR *Listeria* spp. and *E. coli* [63–65].

For *K. pneumoniae*, 65.38% of the isolates were MDR (Fig 4). The highest resistance was observed against ceftriaxone (76.92%), cefotaxime, amoxiclav (61.54%), and trimethoprim-sulfamethoxazole (57.69%). These patterns are consistent with other findings of clinical *K. pneumoniae* resistance to carbapenem [66] and trimethoprim-sulfamethoxazole, ciprofloxacin, and tetracycline [67,68]. For *E. coli*, the highest resistance was observed against amoxicillin and trimethoprim-sulfamethoxazole (81.82%), with comparable levels in food isolates (82–88%) [69] and 100% ampicillin resistance in peri-urban garden isolates from Bangladesh [70]. Despite being rare in our samples, another study from Bangladesh reported *P. aeruginosa* as the most common bacteria in soft beverages and demonstrated the highest resistance to ciprofloxacin (62.5%) [41]. *A. baumannii* showed resistance ranging from 16.7 to 83.3% (Fig 3). The antibiotics from the class of cephalosporins (cefotaxime) and gentamicin showed the highest resistance (83.3%), patterns that were observed in previous studies of clinical isolates [71,72]. According to earlier environmental and dietary MDR findings, *Vibrio* spp. exhibited 10–60% resistance, peaking with amoxicillin (60%) [73,74]. In line with earlier studies of MDR *Staphylococcus* spp. resistant mostly to β-lactams, *S. aureus* (resistance 6.7–80%) and *S. epidermidis* showed the strongest resistance against broad-spectrum medicines such as amoxicillin and cephalosporins [75,76]. According to ambient water investigations [77,78], *Listeria* spp. were 94.44% MDR (Fig 4), with the highest resistance to ampicillin (83.33%), amoxicillin (77.78%), and tetracycline (72.22%).

The MAR index is a straightforward and efficient method for expressing resistance and offers a quantifiable measure of resistance. 31.82% of MAR indices were <0.25, 36.37% between 0.26–0.5, 22.28% between 0.6–0.75, and 9.09% above 0.75. A MAR index >0.2 denotes origin from a high-antibiotic-use environment [79]. Our results are consistent with high MAR values found in bottled water and regional beverages [41,80,81]. Strict hygiene during manufacture and storage is essential since beverages with high MAR indices may act as carriers of resistant microorganisms.

Methicillin resistance in *S. aureus* was assessed by both cefoxitin (30 µg) disk diffusion and *mecA* PCR, in accordance with CLSI 2024 guidelines. Cefoxitin can be used as surrogate marker for *mecA* mediated resistance [29]. Cefoxitin resistance (zone ≤21 mm) was detected in 26.7% of isolates, while *mecA* was identified in 20%. The observed *mecA* prevalence is comparable to reports from food isolates in southwest China (29.5%) [82] and to MRSA from Egyptian and Nigerian food sources [83], suggesting that locally produced electrolyte beverages may act as reservoirs of MRSA contamination via human handling during production. ESBL genes ($bla_{CTX-M}$, $bla_{TEM}$, and $bla_{SHV}$) in *K. pneumoniae* and *E. coli* are consistent with earlier results in poultry farms in the Philippines as well as in isolates of *E. coli* and *K. pneumoniae* from Malaysia and Iraq [84,85]. Historically associated with clinical MDR pathogens, carbapenemase genes ($bla_{NDM}$, $bla_{VIM}$, and $bla_{KPC}$) have been found in food and environmental sources more frequently. The detection of $bla_{NDM}$, and $bla_{KPC}$ in *Enterobacterales* from food samples in Dhaka [86] and in vegetables, water, and hospital and municipal wastewater worldwide [87–89] supports our observation of carbapenemase genes in isolates of Gram-negative beverages and emphasizes the transmission of last-line antibiotic resistance across the food–beverage–environment continuum.

Approximately 40% of isolates were hemolytic (27.27% Gram-positive, 18% Gram-negative), with the majority of *P. aeruginosa, Listeria*, and *Vibrio* isolates exhibiting beta-hemolysis, patterns also noted for *Vibrio* in Singaporean food samples [90,91] and *Listeria* in grass and soil [92]. Surprisingly, *S. epidermidis*, which is normally non-hemolytic, showed

beta-hemolysis; this has been shown in acidic environments, although it varies by strain [93]. According to earlier reports, coagulase positive in 25.08% of samples indicates *S. aureus* contamination, most likely through human interaction during production or packaging [93,94].

One important factor influencing pathogenicity is serum resistance [36]. In our study, 30% of Gram-negative and 25.45% of Gram-positive isolates were resistant to serum treatment. Although Nigerian research discovered that 33.33% of tap water isolates were serum resistant [95], and clinical isolates are generally resistant [96], few studies have looked at serum resistance in beverage isolates. Therefore, more research is necessary to determine the therapeutic significance of serum-resistant beverage isolates.

Of the isolates, almost 70% had the potential to be pathogenic. Meropenem and azithromycin resistance levels were like those of hospital-acquired infections [97], indicating that locally made beverages may have similar resistance characteristics to environmental and clinical reservoirs. The issue is made worse by the fact that these regional goods have not received approval from the Bangladesh Standards and Testing Institution (BSTI) [98].

Since there were no specific microbiological requirements for electrolyte drinks, our findings were compared to non-carbonated beverage standards and bottled water. No coliforms should be found in drinking water, according to WHO water recommendations [99]. Although most samples fulfilled general hygiene standards, several of them had fecal contamination or exceeded acceptable APC values. The FDA, WHO, and BSTI should create specific microbiological guidelines for electrolyte beverages because manufacturers frequently use standards meant for soft drinks or bottled water.

Significant public health consequences result from contaminated electrolyte drinks that include MDR or pathogenic bacteria, especially in developing nations where antibiotic misuse and uneven regulation are common. These findings highlight the necessity of strict quality control in beverage production, including regular microbiological testing, water purification verification, HACCP-based systems, and hygiene training. While there are FDA restrictions for energy drinks [10], electrolyte drinks are not particularly covered by any similar laws. To better understand these drinks as possible reservoirs of resistant bacteria, future studies employing metagenomic and qPCR-based techniques should describe resistance genes and virulence factors.

## Conclusion

This study shows that the microbiological quality of commercially available electrolyte drinks varies, with some samples showing signs of fecal contamination, bacterial loads over permissible limits, and virulent and multidrug-resistant bacteria. Despite the general belief that electrolyte drinks are safe consumer goods, these findings raise questions about their hygienic qualities as well as their possible role as reservoirs for antibiotic-resistant bacteria. One important regulatory gap that must be addressed is the lack of product-specific microbiological standards. Regular monitoring, stringent adherence to hygienic and good manufacturing procedures, and the use of microbiologically safe water sources during production are all necessary to guarantee the microbiological safety of electrolyte drinks.

## Supporting information

**S1 Table. Colony Morphology of Specific Bacteria on Selective Media.** This table lists the kinds of media that are used to grow a certain kind of bacteria, along with the anticipated morphological colonies that will be observed on those media.
(PDF)

**S2 Table. List of all the Antibiotics Used in the Experiment.** It explains the kind of antibiotic group, its potency, spectrum, and interpreting standards.
(PDF)

**S3 Table. Sensitivity profile of all isolated organisms.** The degree of sensitivity exhibited by the bacterial isolates to the antibiotics is shown. The number enclosed by the first bracket is the percentage of isolates.
(PDF)

**S1 Fig. Gram stain of *Klebsiella pneumoniae*.** Here, the bacterial cells are appearing as Gram-negative, short rod-shaped bacilli appearing pink under light microscopy.
(TIF)

**S2 Fig. Gram stain of *Escherichia coli*.** Here, the bacterial cells are appearing as Gram-negative, short rod-shaped bacteria appearing pink under light microscopy.
(TIF)

**S3 Fig. Gram stain of *Pseudomonas aeruginosa*.** Here, the bacterial cells are appearing Gram-negative, rod-shaped cells appearing pink under light microscopy.
(TIF)

**S4 Fig. Gram stain of *Acinetobacter baumannii*.** Here, the bacterial cells are appearing Gram-negative coccobacilli appearing as short rods, stained pink under light microscopy.
(TIF)

**S5 Fig. Gram stain of *Vibrio* spp.** Here, the bacterial cells are appearing Gram-negative, curved rod-shaped cells under light microscopy.
(TIF)

**S6 Fig. Gram stain of *Staphylococcus aureus*.** Here, the bacterial cells are appearing purple-stained Gram-positive cocci arranged in clusters, under light microscopy.
(TIF)

**S7 Fig. Gram stain of *Staphylococcus epidermidis*.** Here, the bacterial cells are appearing Gram-positive cocci appearing as grape-like clusters under light microscopy.
(TIF)

**S8 Fig. Gram stain of *Listeria* spp.** Here, the bacterial cells are appearing purple-stained Gram-positive, short rod-shaped bacteria under light microscopy.
(TIF)

**S9 Fig. Agarose gel electrophoresis of the PCR assay of *Klebsiella pneumoniae* isolates.** Here Lane (M) is a 100 bp DNA marker, and Lane (1–18, 20–22) are some positive samples at 130 bp.
(TIF)

**S10 Fig. Agarose gel electrophoresis of the PCR assay of *Escherichia coli* isolates.** Here Lane (M) is a 100 bp DNA marker, and Lane (1–9) are some positive samples at 585 bp.
(TIF)

**S11 Fig. Agarose gel electrophoresis of the PCR assay of *Pseudomonas aeruginosa* isolates.** Here Lane (M) is a 100 bp DNA marker, and Lane (4, 10, 11, 13–17) are some positive samples at 956 bp.
(TIF)

**S12 Fig. Agarose gel electrophoresis of the PCR assay of *Acinetobacter baumannii* isolates.** Here Lane (M) is a 100 bp DNA marker, and Lane (4, 6–10) are some positive samples at 353 bp.
(TIF)

 

**S13 Fig. Agarose gel electrophoresis of the PCR assay of *Vibrio spp.* isolates.** Here Lane (M) is a 100 bp DNA marker, and Lane (6–11) are some positive samples at ~ (620–689) bp.
(TIF)

**S14 Fig. Agarose gel electrophoresis of the PCR assay of *Staphylococcus aureus* isolates.** Here Lane (M) is a 100 bp DNA marker, and Lane (1–14) are some positive samples at 279 bp.
(TIF)

**S15 Fig. Agarose gel electrophoresis of the PCR assay of *Staphylococcus epidermidis* isolates.** Here Lane (M) is a 100 bp DNA marker, and Lane (1, 3–8) are some positive samples at 503 bp.
(TIF)

**S16 Fig. Agarose gel electrophoresis of the PCR assay of *Listeria spp.* isolates.** Here Lane (M) is a 100 bp DNA marker, and Lane (9, 13, 14, 16–18, 24, 27, 32–33) are some positive samples at 370 bp.
(TIF)

**S17 Fig. Agarose gel electrophoresis of the PCR assay of *mecA* gene.** Here, Lane (M) is a 100 bp DNA marker, and Lane (1–6) are some positive samples at 162 bp.
(TIF)

**S18 Fig. Agarose gel electrophoresis of the PCR assay of *bla*$_{CTX-M}$ gene.** Here, Lane (M) is a 100 bp DNA marker, and Lane (1, 2, 4, 6, 8, 10, 13, 15, 17, 24) are some positive samples at 857 bp.
(TIF)

**S19 Fig. Agarose gel electrophoresis of the PCR assay of *bla*$_{TEM}$ gene.** Here, Lane (M) is a 1k bp DNA marker, and Lane (9–22) are some positive samples at 1080 bp.
(TIF)

**S20 Fig. Agarose gel electrophoresis of the PCR assay of *bla*$_{SHV}$ gene.** Here, Lane (M) is a 100 bp DNA marker and Lane (7–13, 15) are some positive samples at 450 bp.
(TIF)

**S21 Fig. Agarose gel electrophoresis of the PCR assay of *bla*$_{KPC}$ gene.** Here, Lame (M) is a 100 bp DNA marker, and Lane (1–24) are some negative samples that did not give any band at 498 bp.
(TIF)

**S22 Fig. Agarose gel electrophoresis of the PCR assay of *bla*$_{NDM}$ gene.** Here, Lane (M) is a 100 bp DNA marker and Lane (3, 4, 9, 10, 11) are some positive samples at 621 bp.
(TIF)

**S23 Fig. Agarose gel electrophoresis of the PCR assay of *bla*$_{VIM}$ gene.** Here, Lane (M) is a 100 bp DNA marker, and Lane (3, 4, 7, 8, 10, 11, 12, 14, 15) are some positive samples at 501 bp.
(TIF)

**S1 Data. Supplementary Datasheet.**
(XLSX)

## Acknowledgments

The authors would like to sincerely thank all individuals who contributed to the completion of this research by providing invaluable assistance, guidance, and technical support throughout the study.

## Author contributions

**Conceptualization:** Sananda Saha, Fahim Kabir Monjurul Haque.

**Data curation:** Sananda Saha, Ayesha Maliha Khan.

**Formal analysis:** Sananda Saha.

**Funding acquisition:** Fahim Kabir Monjurul Haque.

**Investigation:** Sananda Saha, Ayesha Maliha Khan.

**Methodology:** Sananda Saha, Ayesha Maliha Khan, Fahim Kabir Monjurul Haque.

**Project administration:** Fahim Kabir Monjurul Haque.

**Resources:** Fahim Kabir Monjurul Haque.

**Supervision:** Fahim Kabir Monjurul Haque.

**Visualization:** Sananda Saha.

**Writing – original draft:** Sananda Saha, Fahim Kabir Monjurul Haque.

**Writing – review & editing:** Sananda Saha, Fahim Kabir Monjurul Haque.

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
