## [Decision Letter · Decision Letter 0]

11 Dec 2025

PONE-D-25-58950Microbiological quality assessment of potential pathogenic bacteria and multidrug resistance patterns in commercial electrolyte drinks in Dhaka, BangladeshPLOS One

Dear Dr. Haque,

Thank you for submitting your manuscript to PLOS ONE. After careful consideration, we feel that it has merit but does not fully meet PLOS ONE’s publication criteria as it currently stands. Therefore, we invite you to submit a revised version of the manuscript that addresses the points raised during the review process.

We look forward to receiving your revised manuscript.

Kind regards,

Mengistu Hailemariam Zenebe, PhD

Academic Editor

PLOS One

Journal Requirements:

Reviewers' comments:

Reviewer's Responses to Questions

**Comments to the Author**

1. Is the manuscript technically sound, and do the data support the conclusions?

Reviewer #1: No

Reviewer #2: Yes

2. Has the statistical analysis been performed appropriately and rigorously? 

Reviewer #1: No

Reviewer #2: Yes

3. Have the authors made all data underlying the findings in their manuscript fully available?

Reviewer #1: No

Reviewer #2: Yes

4. Is the manuscript presented in an intelligible fashion and written in standard English?

Reviewer #1: No

Reviewer #2: Yes

5. Review Comments to the Author

Reviewer #1: The authors conducted this study to determine the presence and antimicrobial resistance profiles of various Gram-positive and Gram-negative bacteria isolated from local electrolytes in Bangladesh. This study seems interesting, but I found a lot of serious drawbacks! I believe the authors should check the CLSI guidelines properly before selecting antibiotics against different bacteria. Moreover, I found some serious issues in the AMR results. For example, Klebsiella pneumoniae is intrinsically resistant to ampicillin, amoxicillin, amoxicillin-clavulanic acid, and cefoxitin, but they found 30.77%, 19.23%, 3.85%, and 15.38% of the isolates, respectively. This result doesn’t make sense. The same happened to Pseudomonas aeruginosa, Acinetobacter baumannii, and others. Moreover, you mustn’t follow CLSI 2018 for Vibrio isolates; you should follow CLSI 2015. Also, you used the disk diffusion test for vancomycin, but you mustn’t do that. You must determine MIC values for Staphylococcus aureus against vancomycin to determine their resistance profiles. Overall, I feel the authors didn’t conduct the study properly. They should repeat the study with a proper study design and antimicrobial agent selection. They should also properly evaluate the lab work and results to maintain scientific integrity.

Reviewer #2: The article describe the "Microbiological quality assessment of potential pathogenic bacteria and multidrug

resistance patterns in commercial electrolyte drinks in Dhaka, Bangladesh". The article is well written with sufficient data and the results are convincing. However, the author should consider the following points before finalizing the manuscript.

1. The author isolated a number of bacteria from different samples. the author should add the images of grams staining as well as the characteristic colony of the isolated bacteria in the supplementary materials.

2. It would be good to to show Antimicrobial Resistance (AMR) using heat maps, and this is a widely used and effective method in scientific research and public health surveillance. Heat maps provide a clear visual representation of complex AMR data, allowing researchers and public health officials to identify patterns, trends, and correlations easily.

3. Discuss the production volume and popularity of electrolyte drink in Bangladesh in the introduction section of the manuscript.

4. The authors identified antibiotic resistance only phenotypically, the authors can target known genes responsible for that resistance mechanism.

For Methicillin-resistant Staphylococcus aureus (MRSA): The primary target is the mecA or mecC gene, which encodes penicillin-binding protein 2a (PBP2a), conferring resistance to methicillin and most beta-lactams.

For Extended-Spectrum Beta-Lactamase (ESBL) producers (common in E. coli and Klebsiella): Common genes to screen for include blaCTX-M, blaTEM, and blaSHV subclasses.

For Carbapenem-resistant Gram-negative bacteria: Key carbapenemase genes include blaKPC, blaNDM, blaVIM, blaIMP, and blaOXA-48-like.

For Tetracycline resistance: Common genes are tetA, tetB, tetM, etc.

Data regarding the resistance mechanism will improve the quality of the manuscripts and would be more acceptable to the readers.

6. PLOS authors have the option to publish the peer review history of their article (what does this mean?). If published, this will include your full peer review and any attached files.

Reviewer #1: **Yes:** Md Saiful Islam

Reviewer #2: **Yes:** Sukumar Saha

---

## [Author Response · Author response to Decision Letter 1]

23 Jan 2026

We extend our heartfelt appreciation to the reviewers and editors for their invaluable critiques and suggestions, which have significantly contributed to the enhancement of our manuscript. In response to the reviewers’ feedback, we have carefully revised the manuscript by updating and expanding the reference list, adding new results, including a heatmap illustrating antimicrobial resistance patterns, and incorporating a new figure (heatmap) and an additional table summarizing antimicrobial resistance–related genes. We have prepared a rebuttal letter that responds to each point raised by the academic editor and reviewer(s) and uploaded the letter as a separate file labeled as “Response to Reviewers” along with all other associated files.

---

## [Decision Letter · Decision Letter 1]

28 Apr 2026

PONE-D-25-58950R1Microbiological quality assessment of potential pathogenic bacteria and multidrug resistance patterns in commercial electrolyte drinks in Dhaka, BangladeshPLOS One

Dear Dr. Haque,

Thank you for submitting your manuscript to PLOS ONE. After careful consideration, we feel that it has merit but does not fully meet PLOS ONE’s publication criteria as it currently stands. Therefore, we invite you to submit a revised version of the manuscript that addresses the points raised during the review process.

We look forward to receiving your revised manuscript.

Kind regards,

Mengistu Hailemariam Zenebe, PhD

Academic Editor

PLOS One

Journal Requirements:

Reviewers' comments:

Reviewer's Responses to Questions

**Comments to the Author**

1. If the authors have adequately addressed your comments raised in a previous round of review and you feel that this manuscript is now acceptable for publication, you may indicate that here to bypass the “Comments to the Author” section, enter your conflict of interest statement in the “Confidential to Editor” section, and submit your "Accept" recommendation.

Reviewer #3: (No Response)

Reviewer #4: All comments have been addressed

2. Is the manuscript technically sound, and do the data support the conclusions?

Reviewer #3: Yes

Reviewer #4: Yes

3. Has the statistical analysis been performed appropriately and rigorously? 

Reviewer #3: Yes

Reviewer #4: Yes

4. Have the authors made all data underlying the findings in their manuscript fully available?

Reviewer #3: Yes

Reviewer #4: Yes

5. Is the manuscript presented in an intelligible fashion and written in standard English?

Reviewer #3: Yes

Reviewer #4: Yes

6. Review Comments to the Author

Reviewer #3: (No Response)

Reviewer #4: The authors mostly addressed the raised concnerns. Before the manuscript acceptance, they must carefully revise the italics writing of the scientific name of species within the entire manuscript, including figures and newly inserted references.

7. PLOS authors have the option to publish the peer review history of their article (what does this mean?). If published, this will include your full peer review and any attached files.

Reviewer #3: No

Reviewer #4: No

---

## [Author Response · Author response to Decision Letter 2]

16 May 2026

In response to the reviewers’ feedback, we have carefully revised the manuscript by updating and expanding the reference list, supporting all the relevant data with proper reference in the manuscript, adding all the methodological details and clarifications requested by the reviewers in the Methods section and shortening the introduction and discussion section to fit the structured narrative of a research manuscript. Furthermore, the overall language and clarity of the manuscript have been improved in accordance with the reviewers’ recommendations. We are confident that these revisions have strengthened our manuscript.

---

## [Editor Report · Decision Letter 2]

19 May 2026

Microbiological quality assessment of potential pathogenic bacteria and multidrug resistance patterns in commercial electrolyte drinks in Dhaka, Bangladesh

PONE-D-25-58950R2

Dear Author,

We’re pleased to inform you that your manuscript has been judged scientifically suitable for publication and will be formally accepted for publication once it meets all outstanding technical requirements.

Kind regards,

Mengistu Hailemariam Zenebe, PhD

Academic Editor

PLOS One
---

## [Editor Report · Acceptance letter]

PONE-D-25-58950R2

PLOS One

Dear Dr. Haque,

I'm pleased to inform you that your manuscript has been deemed suitable for publication in PLOS One. Congratulations! Your manuscript is now being handed over to our production team.

Kind regards,

on behalf of

Dr. Mengistu Hailemariam Zenebe

Academic Editor

PLOS One